# Genetic Variants Associated with the Age of Onset Identified by Whole-Exome Sequencing in Fatal Familial Insomnia

**DOI:** 10.3390/cells12162053

**Published:** 2023-08-12

**Authors:** Katrin Thüne, Matthias Schmitz, John Wiedenhöft, Orr Shomroni, Stefan Göbel, Timothy Bunck, Neelam Younas, Saima Zafar, Peter Hermann, Inga Zerr

**Affiliations:** 1Department of Neurology, National Reference Center for Human Spongiform Encephalopathies, University Medical Center, Georg-August University, 37075 Goettingen, Germany; k.thuene@googlemail.com (K.T.); stefan.goebel@med.uni-goettingen.de (S.G.); timothy.bunck@med.uni-goettingen.de (T.B.); neelam.younas@med.uni-goettingen.de (N.Y.); saima.zafar@med.uni-goettingen.de (S.Z.); peter.hermann@med.uni-goettingen.de (P.H.); ingazerr@med.uni-goettingen.de (I.Z.); 2German Center for Neurodegenerative Diseases (DZNE), 37075 Goettingen, Germany; 3Scientific Core Facility Medical Biometry and Statistical Bioinformatics, University Medical Center Goettingen, 37075 Goettingen, Germany; john.wiedenhoeft@med.uni-goettingen.de; 4NGS-Core Unit for Integrative Genomics, Institute of Human Genetics, University Medical Center Goettingen, 37075 Goettingen, Germany; orr.shomroni@med.uni-goettingen.de

**Keywords:** fatal familial insomnia, genetic variants, age of onset, whole-exome sequencing

## Abstract

Fatal familial insomnia (FFI) is a rare autosomal-dominant inherited prion disease with a wide variability in age of onset. Its causes are not known. In the present study, we aimed to analyze genetic risk factors other than the prion protein gene (*PRNP*), in FFI patients with varying ages of onset. Whole-exome sequencing (WES) analysis was performed for twenty-five individuals with FFI (D178N-129M). Gene ontology enrichment analysis was carried out by Reactome to generate hypotheses regarding the biological processes of the identified genes. In the present study, we used a statistical approach tailored to the specifics of the data and identified nineteen potential gene variants with a potential effect on the age of onset. Evidence for potential disease modulatory risk loci was observed in two pseudogenes (*NR1H5P, GNA13P1)* and three protein coding genes *(EXOC1L, SRSF11* and *MSANTD3)*. These genetic variants are absent in FFI patients with early disease onset (19–40 years). The biological function of these genes and *PRNP* is associated with programmed cell death, caspase-mediated cleavage of cytoskeletal proteins and apoptotic cleavage of cellular proteins. In conclusions, our study provided first evidence for the involvement of genetic risk factors additional to *PRNP*, which may influence the onset of clinical symptoms in FFI.

## 1. Introduction

Fatal familial insomnia (FFI) is a very rare and untreatable inherited neurodegenerative prion disease caused by a *PRNP* mutation [1]. Disease hallmarks are progressive sleep loss, rapid progressive dementia, and failure of the autonomic nervous system, followed by death within 6 to 36 months after symptoms begin. FFI is genetically linked to an autosomal-dominant inherited point mutation, D178N, in *PRNP* coupled with methionine (M) at *PRNP* codon 129 in the affected allele of the same gene [2]. FFI patients present high clinical heterogeneity, including a widespread age of disease onset between 36 and 62 years with an average of 51 years, with extremes at 23 and 76 years. In rare cases, clinical symptoms can appear at younger ages [1,3,4]. So far, the polymorphic site at codon 129 in the non-mutated allele of *PRNP* (M or valine (V)) has been reported as the main genetic variant partly modifying neuropathological and clinical features, including age of onset and disease duration [5]. However, the *PRNP* codon 129 genotype alone cannot predict age of onset, and the high level of clinical heterogeneity hampers early identification. FFI patients are often misdiagnosed with other forms of progressive dementias, since their diagnosis requires the presence of a probable prion disease diagnosis in combination with the disease-specific gene mutation [6]. Vulnerable brain regions involved in disease progression are primarily the thalamus, inferior olive and entorhinal cortex, which are affected by severe neuronal loss, astrocytic gliosis and deposition of pathological prion protein (PrP^Sc^) in a regionally dependent manner [1,6,7,8]. Recently, global gene expression patterns analysis in FFI brains yielded impaired fundamental cellular processes, including protein synthesis machinery, transcriptomic regulation and mitochondrial function, as well as activated pathways related to other neurodegenerative diseases such as Alzheimer’s disease and amyotrophic lateral sclerosis [9,10,11]. However, pathogenesis and genetic risk factors other than *PRNP* remain elusive. In recent years, large-scale genome-wide association studies (GWAS) have identified risk genes associated with neurodegenerative diseases [12]. GWAS have been carried out in search of information on human prion diseases, including different types of Creutzfeldt–Jakob disease and Kuru [13,14,15,16], with the most comprehensive study identifying genetic risk factors for sporadic Creutzfeldt–Jakob disease [17]. Up to now, no GWAS-based analyses of disease susceptibility loci have been conducted for FFI.

In the present study, we performed whole-exome sequencing (WES), allowing for high-throughput and unbiased genetic analysis of novel gene discoveries in a cohort of twenty-five FFI patients with variable ages of disease onset in order to detect genetic variants that might be associated with phenotype and disease susceptibility. We aimed to shed light on new genetic factors that may help to predict disease course, to understand the etiology or to improve the diagnosis of FFI.

## 2. Material and Methods

### 2.1. Study Design

#### Patients

In a pilot study, we retrospectively performed a WES analysis of DNA from twenty-five samples obtained from symptomatic FFI patients recruited at the National Reference Center for TSE in Goettingen between the years 2010 and 2021.

The analyzed patient cohort included twenty-five definite FFI cases (12 female, 13 male). Age of disease onset ranged from 19 to 68 years and is listed in detail below (Table 1, Appendix A). All cases are carriers of the D178N mutation in *PRNP* and M/M at *PRNP* codon 129, which was confirmed by sequence analysis. All patients showed typical FFI clinical syndrome, as described previously [18,19].

### 2.2. DNA Extraction from Blood

Genomic DNA was extracted from peripheral blood using a DNeasy blood and tissue kit from Qiagen (Hilden, NRW, Germany). Assessment of genomic DNA quality and integrity was performed via a fragment analyzer indicating non-degraded DNA of good quality (genomic quality number > 9).

### 2.3. Quality Control

We used bcftools 1.12 for consequence annotation, with human genome assembly hg38 and Ensembl annotations version 103. Phasing information was incomplete, and hence removed, so only local consequences could be called. Synonymous markers were excluded from analysis.

### 2.4. Whole-Exome Sequencing

WES analysis of DNA was carried out on twenty-five FFI patients and was performed at the NGS-Core Unit for Integrative Genomics, Institute of Human Genetics, University Medical Center (Goettingen, Germany). The statistical analysis was performed at the Scientific Core Facility Medical Biometry and Statistical Bioinformatics, University Medical Center Goettingen (Goettingen, Germany). To prevent any bias, the analysis was performed in a blinded manner.

### 2.5. Statistical Analysis

We used R 3.6.2 R to analyze the data at three different levels of detail. At the genotype level, binary variables were defined based on whether an individual was homozygous or heterozygous for any given marker (8,987,871 variables). At the marker level, the variable described whether the alternative marker allele was present at least once, regardless of zygosity (7,027,715 variables). At the gene level, we define a binary variable encoding whether an individual had at least one non-synonymous mutation in a given gene or other annotated region, regardless of zygosity and marker type (35,088 variables). For each of these variables, we used RegParallel 1.4.0 [20] to compute a Cox proportional hazards regression model for an age-of-onset outcome predicted by the individual binary variables described above, as well as sex as a covariate. Potential ties were resolved using Breslow’s method, and collinearity was accepted as we expect instances of perfect separation due to the extremely small sample size. Consequently, *p*-values from the likelihood ratio tests are reported. The analysis is by definition exploratory, as no plausible hypothesis for an age-of-onset outcome has yet been formulated. Instead, a vast number of hypotheses are tested indiscriminately. Contrary to popular practice, exploratory screening strategies do not require multiple testing correction (MTC), as the *p*-values provide a sufficient ranking criterion to select candidates for confirmatory analysis [21]. However, we were faced with the conundrum that there is no further cohort available for such confirmatory tests, due to the extreme rarity of the disease. Therefore, multiple testing corrections were advised in this case.

However, this raised another issue. In a setting like ours, many markers will be distributed identically among subjects, which could be due to two factors. On the one hand, identical distribution might be due to synteny, and while only a small subset of markers in such a segment might influence the age of onset, treating each marker separately will likely yield overly conservative MTC and cost us further power in the analysis. In addition, no reliable phasing information could be established from the WES data. Future research might employ long-read whole-genome sequencing to mitigate this problem. Yet, at the current state of affairs, treating identical variables as a group would have the advantage of limiting the number of hypotheses to be tested. 

On the other hand, many markers are expected to be distributed identically simply due to the unfortunate combination of an extremely small sample size and vast number of hypotheses to be tested. In this regard, the exact opposite strategy is advised, and strong correction methods, which can handle positive dependence between *p*-values are required to limit the number of false discoveries, as these cannot be excluded in follow-up experiments for the foreseeable future. We therefore opted for an ad hoc strategy to generate our candidate set. At each level of resolution (gene, marker, genotype), we computed a restricted set of candidates under a stringent MTC criterion both for individual variables with multiplicity, as well as treating identical variables as a single group, and reported the combined result, thereby balancing strong Type I error control with lenient selection criteria. While this is not a canonical approach, we believe it to be justified in this complicated setting. To generate the candidate set in each case, we used a novel MTC approach [22] who showed that closed testing procedures not only provide control of the family-wise error rate, but also yield confidence bounds on the false discovery proportion valid under post hoc selection. In that context, they define an interesting concept called the concentration set (Lemma 4), which excludes all hypotheses for which there is no evidence under the closed testing procedure.

We used the R package Hommel version 1.6 [23] to generate concentration sets both for local Simes tests, as well as Hommel’s robust test, for grouped and individual variables. We reported the union of those sets as candidates since the correlation structure between *p*-values was unclear, as discussed above. Note that this procedure only selects elementary hypotheses, yet the Simes test is non-consonant. Hence, the intersection hypothesis that at least one of many markers affecting a gene underlies the age of onset is not covered by selecting individual markers. Since in non-consonant methods an intersection hypothesis might be rejected while all their constituent hypotheses are accepted, combining markers into a gene-level variable is not superfluous.

### 2.6. Ethical Issues

The study was conducted according to the revised Declaration of Helsinki and Good Clinical Practice guidelines. It has been approved by the local ethic committee at the University Medical Center (Goettingen, Germany) (No. 9/6/0). Informed consent was given by all study participants or their legal next of kin.

## 3. Results

### 3.1. Detection of Genetic Variations in the Exome Associated with a Disease Onset after 40 Years

WES analysis was performed using DNA from twenty-five FFI patients, five patients with a very early onset between 19 and 40 years, and twenty patients with a later disease onset between 42 and 68 years old (Table 1 and Appendix A). We obtained a comprehensive map of genetic variations in the exome of these patients. Subsequently, we defined binary variables regarding non-synonymous mutations on three levels of resolution in the DNA sequencing data and identified a total number of 8,987,871 unique variables at the genotype level; 7,027,715 at the marker level; and 35,088 variables at the gene level. Including sex as a covariate, we performed a time-to-event analysis with subsequent generation of a candidate set based on novel results in multiple-testing correction. Our analysis yielded nineteen disease-modifying gene variants (eight non-protein coding, nine intronic and two in the 3’ untranslated regions (3′-UTR)) in five gene loci present in FFI patients with a disease onset between 42 and 68 years (log (hazard ratio) of −20), but not in early age of onset FFI patients (19–40 years) (Table 2). Among these genes are two pseudogenes, *NR1H5P* and *GNA13P1*, and three protein coding genes, *EXOC1L* (encodes exocyst complex component 1-like), *SRSF11* (encodes serine/arginine-rich splicing factor 1) and *MSANTD3* (encodes Myb/SANT-like DNA-binding domain-containing protein 3) (Figure 1). Moreover, we observed the absence of genetic variants in another pseudogene locus, *ASNSP6* (showed a log (hazard ratio) of −20) in patients with an onset between 19 and 51 years (Figure 1).

### 3.2. Functional Analysis of Disease-Associated Genes

Gene Ontology enrichment analysis was carried out using the online analysis tool Reactome to generate hypotheses regarding the biological processes in which the set of identified genes could potentially be involved. A functional analysis using the identified protein coding genes, *EXOC1L, SRSF11* and *MSANTD3,* together with *PRNP*, the disease-causing gene, revealed enriched gene ontology terms like programmed cell death, caspase-mediated cleavage of cytoskeletal proteins and apoptotic cleavage of cellular proteins (Figure 2).

### 3.3. Analysis of DNA Sequences Data on Marker Level

Finally, we studied the obtained DNA sequence data at the marker level. The most significant non-synonymous mutations with the highest effect inside the FFI cohort are listed in Appendix A.

At the marker level, we identified a number of significant non-synonymous mutations in FFI patients on different chromosomes (Appendix A). According to their distribution pattern, our WES analysis identified the most relevant genetic markers, located on chromosome 8 and 9, which were significantly associated with a certain age/sex in FFI patients (Appendix A).

## 4. Discussion

Recent GWAS studies successfully contributed to the identification of additional genetic risk factors for prion disease other than *PRNP* [13,14,15,17]. Despite not addressing FFI directly, they can provide important insights into prion disease’s pathogenesis. FFI has been described as being under strong genetic control, though less is known regarding risk factors besides *PRNP*. In affected families, twin studies suggested that identical *PRNP* codon 129 MV genotypes can also exhibit some degree of phenotypic heterogeneity [5]. 

In the present study, our main objective was an explorative analysis to identify genetic variants that might be associated with the age of onset in FFI. In twenty-five individuals with FFI with different ages of onset, exhibiting M/M at *PRNP* codon 129, we performed a WES analysis and used a statistical screening approach tailored to the peculiarities of the data.

We identified non-*PRNP* genetic variants in two pseudogenes *NR1H5P* and *GNA13P1*, as well as in the protein coding genes *EXOC1L*, *SRSF11* and *MSANTD3*, as potential modifiers. These genetic variants were absent in FFI patients with very early ages of onset below 41 years.

Genetic variants in another pseudogene locus, *ASNSP6*, were observed to be absent in patients with an onset between 19 and 51 years old.

The identified candidates were further studied via Gene Ontology enrichment analysis, suggesting their role in neurodegenerative mechanisms. Most of those gene variants seemed to occur in a set of patients with an age of onset older than 40 years, suggesting a protective contribution to mechanisms modifying disease susceptibility and phenotypic heterogeneity. The identified risk gene loci are expressed in various human tissue types, including the brain. Although less is known regarding the physiological protein functions yet, they are suggested to play a role in fundamental cellular processes. 

Most of the genes encode proteins with multiple functions. For example, the protein coding gene *NR1H5* (corresponds to the pseudogene *NR1H5P*) is also known as liver X receptor beta (LXRβ). It belongs to the nuclear receptor superfamily and plays a crucial role in lipid and cholesterol metabolism. The protein is also a potential target for atherosclerosis treatment [24]. 

A potential association with FFI may be the finding that the knock-out of *NR1H5* in mice may influence the synaptic transmission in the prefrontal cortex [25]. Moreover, an activation of the hippocampal LXRβ improves sleep-deprived cognitive impairment by inhibiting neuroinflammation [26].

The functions of other genes can be partially linked to FFI, such as the involvement of *SRSF11*, in pre-mRNA splicing [27,28] and *MSANTD3*, involved in DNA binding and regulation of gene expression and protein synthesis [29]. Moreover, both genes are reported to play a role in cancer [29,30]. In this context, PrP^C^, the physiological counterpart of PrP^Sc^ in prion diseases, is also involved in cancer by influencing cell survival and the invasion/metastasis of cancer cells, suggesting a connection between both diseases [31].

*EXOC1L* is predicted to be involved in membrane transport processes from the Golgi apparatus to the plasma membrane, as well as in exocytosis (Alliance of Genome Resources, 2021).

The detection of genetic variants in pseudogenes (NR1H5P and GNA13P1) raised the question of the role of pseudogenes as disease modifiers. Pseudogenes do not have protein coding capability; however, they may play a role in epigenetic regulation, gene transcription and post-transcriptional gene regulation. Pseudogenes are reported to play a role in the development and clinical manifestation of prion diseases and related neurodegenerative diseases [32,33].

However, the exact role of *GNA13P1*, which derives from the *GNA13* gene, in FFI, remains unclear. *GNA13* is an important member of the G protein alpha subunit family and may play a role in intracellular signaling pathways [34].

Another study [17] identified two other non-*PRNP* gene candidates, *STX6* and *GAL3ST1*, which are associated with the risk of developing a sporadic Creutzfeldt–Jakob disease. *GAL3ST1* is likely related to common variants that alter the protein sequence, whereas risk variants in *STX6* associate with increased expression of the major transcripts in disease-relevant brain regions. In our FFI cohort, we could not observe variations in those gene loci, which may be due to the study design and the composition of the patient cohorts. Jones and colleagues applied a heterogeneous study cohort of 5208 cases from numerous countries/regions which were distributed across a two-stage study design [14]: 4110 cases were genotyped using the Illumina Omniexpress array in the discovery phase, and an additional 1098 cases were genotyped at the lead variant in each hit locus using minor groove binding probes in the replication phase. Control data were obtained from publicly available datasets matched for country. In addition, they were mainly looking for risk genes for the development of a sporadic form of a prion disease [17]. We followed a completely different strategy, as our German cohort consists of homogenous FFI cases (all with the *PRNP* codon 129 MM genotype, carrying the D178N mutation), and we aimed to identify risk loci which may influence the age of onset of FFI.

A limitation of our study is the small number of FFI cases in particular with an early onset, which is due to the rarity of the disease. This is the first pilot study on this topic, and the data would have to be confirmed in a larger FFI patient cohort. 

In summary, we collected evidence for the presence of genetic variants other than the *PRNP* gene that modulate or are at least associated with disease onset. We believe that these data might help us to understand the disease ethology and might be useful to predict the course or even the onset of FFI.

## 5. Conclusions

Fatal familial insomnia (FFI) belongs to the group of genetic prion diseases and is caused by an autosomal-dominant inherited point mutation, D178N, in *PRNP*. The course and the duration of FFI is very heterogenic and cannot be sufficiently explained so far. Due to the extreme rarity of the disease, studies on genetic risk factors or loci influencing the course of FFI are very challenging. 

Nineteen disease-modifying gene variants (eight non-protein coding, nine intronic and two in the 3′ untranslated regions (3′-UTR)) present in FFI patients with a disease onset between 42 and 68 years, but not in early onset FFI patients (19–40 years), were identified, indicating an association of these variants with the onset of FFI.

The five genes (*NR1H5P*, *GNA13P1*, *EXOC1L*, *SRSF11* and *MSANTD3*) and *PRNP* are involved in physiological processes, such as programmed cell death, caspase-mediated cleavage of cytoskeletal proteins and apoptotic cleavage of cellular proteins, which might influence the course of FFI.

## Figures and Tables

**Figure 1 cells-12-02053-f001:**
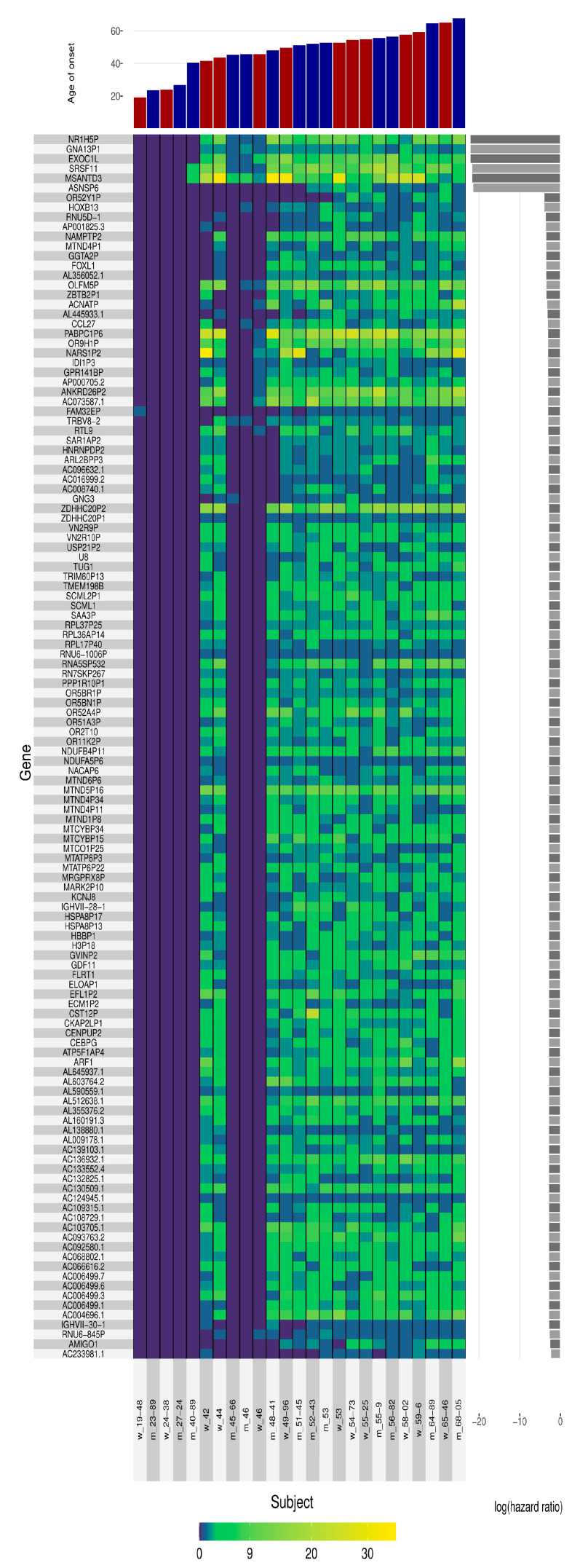
Identification of gene loci associated with an early disease onset in FFI. WES analysis from 25 FFI patients resulted in the identification of gene and pseudo-gene loci, which are associated with different ages of disease onset. Lack of genetic variants in *NR1H5P*, *GNA13P1*, *EXOC1L*, *SRSF11*, and *MSANTD3* gene loci were observed in FFI patients with an onset between 19 and 40 years indicated by the low log (hazard ratio). Age of onset (top panel) genes were ranked with respect to hazard ratio (middle panel), and analyzed subjects (lower panel) sorted by top affected genes. The color of heatmap reflects the number of variants per gene. Red and blue bars indicate female and male subjects (top panel); grey bars indicate the log/hazard ratio.

**Figure 2 cells-12-02053-f002:**
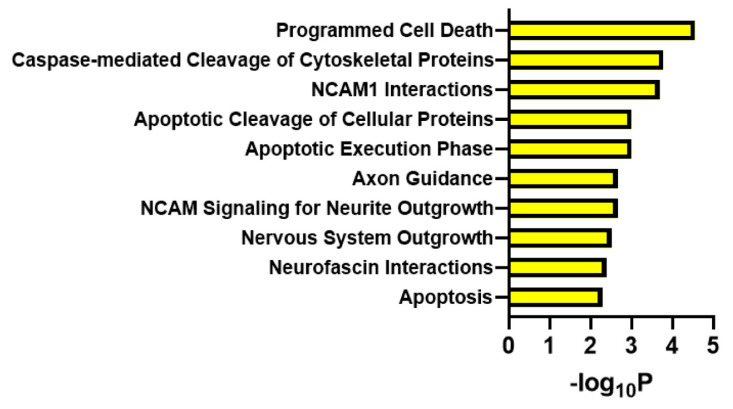
Physiological functions of disease-onset-associated genes and *PRNP*. A functional analysis using *EXOC1L*, *SRSF11*, *MSANTD3* and *PRNP* was carried out by using the analysis tool Reactome (European Bioinformatics Institute, Hinxton, United Kingdom), indicating an involvement of these genes in neurodegenerative processes.

**Table 1 cells-12-02053-t001:** Overview on the study cohort consisting of 25 samples of FFI patients.

FFI Cohort	Early Onset: 19–40 Years	Late Onset: 42–68 Years
Gender (F/M)	2/3	10/10
Age ± SD	27 ± 13	53 ± 15

**Table 2 cells-12-02053-t002:** Identification of alternative allele sequences within the 5 genes associated with a later disease onset than 40 years. Within the candidate gene set, we identified by effect size several single nucleotide variations within these genes associated with a disease onset after 40 years. CHR: chromosome; SNP: single nucleotide polymorphism; BP: base pairs; REF/ALT: reference/alternative allele; NMD: nonsense-mediated mRNA decay; m: male; f: female; utr, untranslated regions.

Gene	CHR	SNP ID	BP	REF	ALT	Location	Patient Age/Sex (m/f)
*NR1H5P*	1	rs360599	114845377	G	T	Non protein coding	48 m, 51 m, 53 m, 64 m, 68 m, 44 f, 53 f, 55 m, 65 f
		rs360607	114849780	C	A		48 m, 51 m, 52 m 53 m, 64 m, 68 m 44 f, 49 f, 53 f, 54 f, 55 f
		rs360608	114850093	C	T		48 m, 52 m, 42 f, 44 f, 49 f, 55 f, 51 m, 53 m, 55 m, 64 m, 53 f, 54 f, 59 f
		rs75631798	114851252	G	T		48 m, 55 m, 52 m, 53 m, 64 m
		rs360609	114851255	C	T		48 m, 55 m, 44 f, 49 f, 54 f, 52 m, 53 m, 64 m
		rs360610	114851294	T	C		48 m, 53 m, 64 m 68 m, 44 f, 49 f, 53 f, 59 f, 54 f
*GNA13P1*	2	rs1594400	79574542	A	G		46 m, 48 m, 52 m, 53 m, 68 m, 42 f, 46 f, 49 f, 53 f, 55 f, 59 f, 65 f, 51 m, 55 m, 44 f, 54 f, 58 f
		rs2685151	79574796	G	A		45 m, 46 m,48 m, 51 m, 52 m, 53 m, 68 m, 42 f, 49 f, 55 f, 59 f, 56 m, 64 m, 54 f, 58 f
*EXOC1L*	4	rs1447039	55828917	G	C	Intronic (NMD)	48 m, 51 m, 55 m, 42 f, 46 f, 49 f, 53 f, 54 f, 55 f, 59 f, 65 f
		rs1992813	55832233	T	A		45 m, 48 m, 52 m, 55 m, 56 m, 64 m, 68 m, 44 f, 49 f, 53 f, 54 f, 55 f, 58 f, 59 f, 65 f
		rs1447049	55836725	G	A		48 m, 51 m, 52 m, 53 m, 55 m, 64 m, 68 m, 42 f, 44 f, 49 f,54 f, 55 f, 58 f, 59 f
*SRSF11*	1	rs647872	70220006	G	A	Intronic (processed transcr.)	46 m, 48 m, 51 m, 52 m, 55 m, 64 m, 68 m, 42 f, 44 f, 49 f, 53 f, 58 f, 65 f, 53 m, 54 f
		rs169172	70223994	A	G	Intronic (NMD)	48 m, 51 m, 53 m, 55 m, 64 m, 42 f, 44 f, 49 f, 55 f, 58 f, 65 f
		rs10661326	70252590	A	ATT	3_prime_utr	52 m, 55 m, 56 m, 64 m, 68 m, 42 f, 44 f, 49 f, 54 f, 65 f
*MSANTD3*	9	rs57241635	100430270	G	A	Intronic (protein coding)	46 m, 48 m, 56 m, 44 f, 49 f, 59 f,
		rs60353063	100434542	G	GA		48 m, 44 f, 49 f, 58 f, 59 f, 46 m
		rs10114432	100443395	A	G	Intronic (processed transcr.)	51 m, 55 m, 68 m, 44 f, 49 f, 55 f
		rs73655549	100445372	G	A		46 m, 48 m, 56 m, 44 f, 49 f, 53 f,58 f, 59 f
		rs1556487	100451665	A	G	3_prime_utr	48 m, 52 m, 64 m, 53 f, 55 f, 59 f

## Data Availability

Data are contained within the article or Appendix A.

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
