# Peer review of "Genetic Variants Associated with the Age of Onset Identified by Whole-Exome Sequencing in Fatal Familial Insomnia"

_cells, 2023, doi:10.3390/cells12162053_

Round 1

Reviewer 1 Report

This interesting manuscript describes a study aimed at sequencing the whole exomes of 25 patients suffering Fatal Familial Insomnia. The objective is to identify genes that might play a role in the intriguing disparity of age at onset of the disease. It should be noted that while the ultimate casue of such clinical onset is the specific, dominant FFI mutation in the prion protein gene (PRNP), clinical disease can manifest between 19 and 63 years of age. Identifying  the molecular underpinnings of such phenomenon is of key relevance for two reasons: 1) it will shed light on the molecular pathogenic mechanisms of prion diseases, which are poorly understood to date; 2) It might identify targets of therapeutic interventions. Therefore this istudy is very timely. It should be also noted that while whole exome sequencing is now relatively routine, the bottleneck for these kind of studies as applied to prion diseases is the rarity of the genetic forms of the disease, such as FFI. Therefore the Authors´ access to the German Reference Center for Transmissible Spongiform Encephalopathies, and their careful selection, clinical characterization and sample collection is an invaluable and rare asset.

In this context, I have little to say about the analytical methodology, which appears straightforward, utilizing state of the art biochemical and bioinformatic methods and approaches. The study yielded 19 disease modifying gene variants  in five gene loci present in FFI patients with an "older" disease onset (42-68 years), but not in "earlier" onset FFI patients (19-40 years). Two of these are pseudogenes, NR1H5P and GNA13P1 and three protein-coding genes: EXOC1L, SRSF11 and MSANTD3. The protein coding genes, although not very well known, seem to be involved in a variaty of potentially relevant patthways such as apoptosis or protein metabolism. 

I have just very few suggestions that in my opinion might improve the manuscript:

1) As a reader, I was a bit disappointed not to find more information about the identified genes and their roles. I appreciate the Authors´ restraint in avoiding speculation in their Discussion; it is indeed too early to speculate too much. But at least a more detailed summary of what is known about these genes, and perhaps a cautious hypothesis on how they might impinge on the pathology of FFI would be wellcome. Thus, at least two of the genes, SRSF11 and MSANTD3, have beeen involved in cancer; what might this mean? Is there a wider connection between cancer and prion diseases?

2) The title is a bit ackward, written in the past tense as it is. I would suggest re-writing it in the passive voice: "Risk gene-variants associated with the age at onset identified by...". 

Author Response

Dear Editor, Dear Reviewers,

We would like to thank you very much for the feedback on our manuscript and are very grateful to the reviewers for their helpful comments and suggestions. We have revised the manuscript and addressed all the points raised upon review. Please find our responses to the reviewer’s comments below.

Academic Editor Notes:

I cordially invite you to revise the manuscript and address the comments put forth by the reviewers. Specifically, please provide detailed information regarding the identified genes as suggested by reviewer #1. Additionally, I kindly request that you enhance the discussion pertaining to GAL3ST1 and STX6, which were identified in sCJD, based on the insightful comments provided by reviewer #2.

Response:

We would like to thank you for your comment and the kind possibility to submit a revised version. We enclosed a detailed information regarding the identified genes (see the response for Reviewer 1). Moreover, we extended the discussion part and added a paragraph about potential reason why we could not find GAL3ST1 and STX6 as risk genes in our FFI cohort. Please consider the response to reviewer 2.

Reviewer 1

This interesting manuscript describes a study aimed at sequencing the whole exomes of 25 patients suffering Fatal Familial Insomnia. The objective is to identify genes that might play a role in the intriguing disparity of age at onset of the disease. It should be noted that while the ultimate casue of such clinical onset is the specific, dominant FFI mutation in the prion protein gene (PRNP), clinical disease can manifest between 19 and 63 years of age. Identifying  the molecular underpinnings of such phenomenon is of key relevance for two reasons: 1) it will shed light on the molecular pathogenic mechanisms of prion diseases, which are poorly understood to date; 2) It might identify targets of therapeutic interventions. Therefore this istudy is very timely. It should be also noted that while whole exome sequencing is now relatively routine, the bottleneck for these kind of studies as applied to prion diseases is the rarity of the genetic forms of the disease, such as FFI. Therefore the Authors´ access to the German Reference Center for Transmissible Spongiform Encephalopathies, and their careful selection, clinical characterization and sample collection is an invaluable and rare asset.

In this context, I have little to say about the analytical methodology, which appears straightforward, utilizing state of the art biochemical and bioinformatic methods and approaches. The study yielded 19 disease modifying gene variants in five gene loci present in FFI patients with an "older" disease onset (42-68 years), but not in "earlier" onset FFI patients (19-40 years). Two of these are pseudogenes, NR1H5P and GNA13P1 and three protein-coding genes: EXOC1L, SRSF11 and MSANTD3.

Response:

Thank you very much; we really appreciate your comments.

Point 1) The protein coding genes, although not very well known, seem to be involved in a variety of potentially relevant pathways such as apoptosis or protein metabolism. I have just very few suggestions that in my opinion might improve the manuscript: As a reader, I was a bit disappointed not to find more information about the identified genes and their roles. I appreciate the Authors´ restraint in avoiding speculation in their Discussion; it is indeed too early to speculate too much. But at least a more detailed summary of what is known about these genes, and perhaps a cautious hypothesis on how they might impinge on the pathology of FFI would be welcome. Thus, at least two of the genes, SRSF11 and MSANTD3, have beeen involved in cancer; what might this mean? Is there a wider connection between cancer and prion diseases?

Response:

We agree to the suggestion of the reviewer and revised our discussion by adding the following paragraph about the physiological function of the newly identified genes:

The physiological function of the five identified genes is not well described in literature yet, making a direct association with FFI challenging.

Most of the genes encode proteins with multiple functions. For examples, the protein-coding gene NR1H5 also known as liver X receptor beta (LXRβ). It belongs to the nuclear receptor superfamily and plays a crucial role in lipid and cholesterol metabolism. The protein is a potential target for atherosclerosis treatment (Salvia et al., 2022).

A potential association with FFI may be the finding that knock-out of NR1H5 in mice may influence the synaptic transmission in the prefrontal cortex (Li et al., 2021). Moreover, an activation of the hippocampal LXRβ improves sleep-deprived cognitive impairment by inhibiting neuroinflammation (Chen et al., 2021).

The functions of other genes can be partially linked to FFI, such as the involvement of SRSF1l, in pre-mRNA splicing (Herdlevaer et al., 2020; Wang et al. 2022)  and MSANTD3, involved in DNA binding and regulation of gene expression and protein synthesis (Barasch et al. 2017). Moreover, both gene are reported to play a role in cancer (Scorilas et al., 2001; Barasch et al., 2017) (as mentioned by the reviewer). In this context, PrPC, the physiological counterpart of PrPSc in prion disease, is also involved in cancer by influencing the cell survival, invasion/metastasis of cancer cells suggestion a connection between both diseases (Ding et al., 2021; Frontiers; Armocida et al., 2023).

EXOC1L is predicted to be involved in membrane transport processes from the Golgi apparatus to the plasma membrane as well as in exocytosis (Alliance of Genome Resources, 2021).

Two gene loci associated to a later disease onset were detected in the pseudogene GNA13P1. Pseudogenes do not have protein-coding capability, however, they were considered to play a role in epigenetic regulation, gene transcription, and post transcriptional gene regulation. Pseudogenes are discussed to play a role in the development and clinical manifestation of prion diseases and related neurodegenerative diseases (Brayton et al. 2004; Liu et al. 2021). The exact role of GNA13P1, which derives from the GNA13 gene (Teo et al. 2016), in FFI remained unclear.  GNA13 is an important member of the G protein alpha subunit family and may play a role in intracellular signaling pathways (line 243-271).

Point 2) The title is a bit ackward, written in the past tense as it is. I would suggest re-writing it in the passive voice: "Risk gene-variants associated with the age at onset identified by...". 

Response: We revised the title of the manuscript as suggested to “Risk gene-variants associated with the age at onset identified by Whole-exome sequencing in Fatal Familial Insomnia”.

With these answers and modifications in our revised review, we hope to have adequately addressed all points raised upon review and hope that our manuscript is now acceptable for publication.

Sincerely yours,

Matthias Schmitz

References:

Barasch N, Gong X, Kwei KA, Varma S, Biscocho J, Qu K, Xiao N, Lipsick JS, Pelham RJ, West RB, Pollack JR. Recurrent rearrangements of the Myb/SANT-like DNA-binding domain containing 3 gene (MSANTD3) in salivary gland acinic cell carcinoma. PLoS One. 2017 Feb 17;12(2):e0171265. doi: 10.1371/journal.pone.0171265. PMID: 28212443; PMCID: PMC5315303.

Herdlevaer I, Kråkenes T, Schubert M, Vedeler CA. Localization of CDR2L and CDR2 in paraneoplastic cerebellar degeneration. Ann Clin Transl Neurol. 2020 Nov;7(11):2231-2242. doi: 10.1002/acn3.51212. Epub 2020 Oct 3. PMID: 33009713; PMCID: PMC7664253.

Jones E, Hummerich H, Viré E, Uphill J, Dimitriadis A, Speedy H, Campbell T, Norsworthy P, Quinn L, Whitfield J, Linehan J, Jaunmuktane Z, Brandner S, Jat P, Nihat A, How Mok T, Ahmed P, Collins S, Stehmann C, Sarros S, Kovacs GG, Geschwind MD, Golubjatnikov A, Frontzek K, Budka H, Aguzzi A, Karamujić-ÄŒomić H, van der Lee SJ, Ibrahim-Verbaas CA, van Duijn CM, Sikorska B, Golanska E, Liberski PP, Calero M, Calero O, Sanchez-Juan P, Salas A, Martinón-Torres F, Bouaziz-Amar E, Haïk S, Laplanche JL, Brandel JP, Amouyel P, Lambert JC, Parchi P, Bartoletti-Stella A, Capellari S, Poleggi A, Ladogana A, Pocchiari M, Aneli S, Matullo G, Knight R, Zafar S, Zerr I, Booth S, Coulthart MB, Jansen GH, Glisic K, Blevins J, Gambetti P, Safar J, Appleby B, Collinge J, Mead S. Identification of novel risk loci and causal insights for sporadic Creutzfeldt-Jakob disease: a genome-wide association study. Lancet Neurol. 2020 Oct;19(10):840-848. doi: 10.1016/S1474-4422(20)30273-8. Epub 2020 Sep 16. PMID: 32949544; PMCID: PMC8220892.

Li X, Zhong H, Wang Z, Xiao R, Antonson P, Liu T, Wu C, Zou J, Wang L, Nalvarte I, Xu H, Warner M, Gustafsson JA, Fan X. Loss of liver X receptor β in astrocytes leads to anxiety-like behaviors via regulating synaptic transmission in the medial prefrontal cortex in mice. Mol Psychiatry. 2021 Nov;26(11):6380-6393. doi: 10.1038/s41380-021-01139-5. Epub 2021 May 7. PMID: 33963286.

Matsumoto S, Okada J, Yamada E, Saito T, Okada K, Watanabe T, Nakajima Y, Ozawa A, Okada S, Yamada M. Overexpressed exocyst complex component 3-like 1 spontaneously induces apoptosis. Biomed Res. 2021;42(3):109-113. doi: 10.2220/biomedres.42.109. PMID: 34092752.

Qiu C, Wang M, Yu W, Rong Z, Zheng HS, Sun T, Liu SB, Zhao MG, Wu YM. Activation of the Hippocampal LXRβ Improves Sleep-Deprived Cognitive Impairment by Inhibiting Neuroinflammation. Mol Neurobiol. 2021 Oct;58(10):5272-5288. doi: 10.1007/s12035-021-02446-2. Epub 2021 Jul 19. PMID: 34278533.

Savla SR, Prabhavalkar KS, Bhatt LK. Liver X receptor: a potential target in the treatment of atherosclerosis. Expert Opin Ther Targets. 2022 Jul;26(7):645-658. doi: 10.1080/14728222.2022.2117610. Epub 2022 Sep 5. PMID: 36003057.

Scorilas A, Kyriakopoulou L, Katsaros D, Diamandis EP. Cloning of a gene (SR-A1), encoding for a new member of the human Ser/Arg-rich family of pre-mRNA splicing factors: overexpression in aggressive ovarian cancer. Br J Cancer. 2001 Jul 20;85(2):190-8. doi: 10.1054/bjoc.2001.1885. PMID: 11461075; PMCID: PMC2364031.

Barasch et al., 2017

Teo CR, Casey PJ, Rasheed SA. The GNA13-RhoA signaling axis suppresses expression of tumor protective Kallikreins. Cell Signal. 2016 Oct;28(10):1479-88. doi: 10.1016/j.cellsig.2016.07.001. Epub 2016 Jul 14. PMID: 27424208.

Armocida D, Busceti CL, Biagioni F, Fornai F, Frati A. The Role of Cellular Prion Protein in Glioma Tumorigenesis Could Be through the Autophagic Mechanisms: A Narrative Review.

Int J Mol Sci. 2023 Jan 11;24(2):1405. doi: 10.3390/ijms24021405.

Ding M, Chen Y, Lang Y, Cui L. (2021) The Role of Cellular Prion Protein in Cancer Biology: A Potential Therapeutic Target. Front Oncol. 2021 Sep 14;11:742949. doi: 0.3389/fonc.2021.742949. eCollection 2021.PMID: 34595121

Reviewer 2 Report

I disagree with the authors contention that "strong evidence for disease modulatory risk loci was observed in five genes." I disagree because the experiments are not adequately powered given the multiple correction issues. 

The observed associations are likely spurious. For consideration, Jones et al., (PMID:32949544) examined 5208 cases of CJD vs health controls and found only two non-prnp candidates STX6 and GAL3ST1. In Thune et al, 5 candidates are found, and the contrast is between early onset and late onset.

What is happening with GAL3ST1 and STX6 in FFI? Perhaps these deserve a special look?

Do these variants affect other prion disease or have any plausible role in the age of onset of CJD? What is the rationale for FFI being a "special" prion disease that then has its own distinct set of extra-prnp disease modifying alleles?

I am not opposed to the publication of this manuscript based upon the lack of adequate power for a rare disease such as FFI. The authors should be allowed to present it as a negative result while still discussing the potential that the observations are real.

no issues

Author Response

 Dear Reviewers, 

We would like to thank you very much for the feedback on our manuscript and are very grateful to the reviewers for their helpful comments and suggestions. We have revised the manuscript and addressed all the points raised upon review. Please find our responses to the reviewer’s comments below.

Reviewer 2

Point 1) I disagree with the authors contention that "strong evidence for disease modulatory risk loci was observed in five genes." I disagree because the experiments are not adequately powered given the multiple correction issues. The observed associations are likely spurious. For consideration, Jones et al., (PMID:32949544) examined 5208 cases of CJD vs health controls and found only two non-prnp candidates STX6 and GAL3ST1. In Thune et al, 5 candidates are found, and the contrast is between early onset and late onset.

What is happening with GAL3ST1 and STX6 in FFI? Perhaps these deserve a special look?

Response:

We agree with the reviewer that, by necessity, the study is underpowered for the amount of statistical tests to be performed, and that multiple testing is a concern. We discuss this in the 'Statistical Analysis' section of the paper, along with other particularities of the problem at hand. There, we argue that it is precisely for those reasons that we opted for a new and unconventional approach to address the MTC and other issues in order to carefully minimize spurious results. We therefore respectfully disagree with the reviewer's assessment. However, we do agree that 'strong evidence' might be an overstatement, given that our approach is highly non-standard and not well-established, and have changed “strong evidence….” to “evidence for disease modulatory risk loci was observed in five genes….”, (line 42).

Secondly, we revised our discussion part and added following paragraph:

The study from Jones et al., 2020 identified two other non-prnp gene candidates, STX6 and GAL3ST1, which are associated with the risk to develop a sporadic Creutzfeldt-Jakob disease (sCJD). GAL3ST1 is likely related to common variants that alter the protein sequence, whereas risk variants in STX6 associate with increased expression of the major transcripts in disease-relevant brain regions. In our FFI patient cohort we could not observe variations in those gene loci, which may depend on the study design and on the composition of the patient cohorts. Jones and colleague applied a heterogeneous study cohort of 5208 cases from numerous countries/regions which were distributed across a two-stage study design: 4110 cases were genotyped using Illumina Omniexpress array in the discovery phase and an additional 1098 cases were genotyped at the lead variant in each hit locus using minor groove binding probes in the replication phase. Control data was obtained from publicly available datasets matched for country. In addition, they were mainly looking for risk genes for development of a sporadic form of a prion disease. We followed a completely different strategy, our German cohort consist of homogenous FFI cases (all with the codon 129 MM genotype, carrying the D178N mutation) and we aimed to identify risk loci which may influence the age at onset of FFI (line 273-288).

With these answers and modifications in our revised review, we hope to have adequately addressed all points raised upon review and hope that our manuscript is now acceptable for publication.

Sincerely yours,

Matthias Schmitz

References:

Barasch N, Gong X, Kwei KA, Varma S, Biscocho J, Qu K, Xiao N, Lipsick JS, Pelham RJ, West RB, Pollack JR. Recurrent rearrangements of the Myb/SANT-like DNA-binding domain containing 3 gene (MSANTD3) in salivary gland acinic cell carcinoma. PLoS One. 2017 Feb 17;12(2):e0171265. doi: 10.1371/journal.pone.0171265. PMID: 28212443; PMCID: PMC5315303.

Herdlevaer I, Kråkenes T, Schubert M, Vedeler CA. Localization of CDR2L and CDR2 in paraneoplastic cerebellar degeneration. Ann Clin Transl Neurol. 2020 Nov;7(11):2231-2242. doi: 10.1002/acn3.51212. Epub 2020 Oct 3. PMID: 33009713; PMCID: PMC7664253.

Jones E, Hummerich H, Viré E, Uphill J, Dimitriadis A, Speedy H, Campbell T, Norsworthy P, Quinn L, Whitfield J, Linehan J, Jaunmuktane Z, Brandner S, Jat P, Nihat A, How Mok T, Ahmed P, Collins S, Stehmann C, Sarros S, Kovacs GG, Geschwind MD, Golubjatnikov A, Frontzek K, Budka H, Aguzzi A, Karamujić-ÄŒomić H, van der Lee SJ, Ibrahim-Verbaas CA, van Duijn CM, Sikorska B, Golanska E, Liberski PP, Calero M, Calero O, Sanchez-Juan P, Salas A, Martinón-Torres F, Bouaziz-Amar E, Haïk S, Laplanche JL, Brandel JP, Amouyel P, Lambert JC, Parchi P, Bartoletti-Stella A, Capellari S, Poleggi A, Ladogana A, Pocchiari M, Aneli S, Matullo G, Knight R, Zafar S, Zerr I, Booth S, Coulthart MB, Jansen GH, Glisic K, Blevins J, Gambetti P, Safar J, Appleby B, Collinge J, Mead S. Identification of novel risk loci and causal insights for sporadic Creutzfeldt-Jakob disease: a genome-wide association study. Lancet Neurol. 2020 Oct;19(10):840-848. doi: 10.1016/S1474-4422(20)30273-8. Epub 2020 Sep 16. PMID: 32949544; PMCID: PMC8220892.

Li X, Zhong H, Wang Z, Xiao R, Antonson P, Liu T, Wu C, Zou J, Wang L, Nalvarte I, Xu H, Warner M, Gustafsson JA, Fan X. Loss of liver X receptor β in astrocytes leads to anxiety-like behaviors via regulating synaptic transmission in the medial prefrontal cortex in mice. Mol Psychiatry. 2021 Nov;26(11):6380-6393. doi: 10.1038/s41380-021-01139-5. Epub 2021 May 7. PMID: 33963286.

Matsumoto S, Okada J, Yamada E, Saito T, Okada K, Watanabe T, Nakajima Y, Ozawa A, Okada S, Yamada M. Overexpressed exocyst complex component 3-like 1 spontaneously induces apoptosis. Biomed Res. 2021;42(3):109-113. doi: 10.2220/biomedres.42.109. PMID: 34092752.

Qiu C, Wang M, Yu W, Rong Z, Zheng HS, Sun T, Liu SB, Zhao MG, Wu YM. Activation of the Hippocampal LXRβ Improves Sleep-Deprived Cognitive Impairment by Inhibiting Neuroinflammation. Mol Neurobiol. 2021 Oct;58(10):5272-5288. doi: 10.1007/s12035-021-02446-2. Epub 2021 Jul 19. PMID: 34278533.

Savla SR, Prabhavalkar KS, Bhatt LK. Liver X receptor: a potential target in the treatment of atherosclerosis. Expert Opin Ther Targets. 2022 Jul;26(7):645-658. doi: 10.1080/14728222.2022.2117610. Epub 2022 Sep 5. PMID: 36003057.

Scorilas A, Kyriakopoulou L, Katsaros D, Diamandis EP. Cloning of a gene (SR-A1), encoding for a new member of the human Ser/Arg-rich family of pre-mRNA splicing factors: overexpression in aggressive ovarian cancer. Br J Cancer. 2001 Jul 20;85(2):190-8. doi: 10.1054/bjoc.2001.1885. PMID: 11461075; PMCID: PMC2364031.

Barasch et al., 2017

Teo CR, Casey PJ, Rasheed SA. The GNA13-RhoA signaling axis suppresses expression of tumor protective Kallikreins. Cell Signal. 2016 Oct;28(10):1479-88. doi: 10.1016/j.cellsig.2016.07.001. Epub 2016 Jul 14. PMID: 27424208.

Armocida D, Busceti CL, Biagioni F, Fornai F, Frati A. The Role of Cellular Prion Protein in Glioma Tumorigenesis Could Be through the Autophagic Mechanisms: A Narrative Review.

Int J Mol Sci. 2023 Jan 11;24(2):1405. doi: 10.3390/ijms24021405.

Ding M, Chen Y, Lang Y, Cui L. (2021) The Role of Cellular Prion Protein in Cancer Biology: A Potential Therapeutic Target. Front Oncol. 2021 Sep 14;11:742949. doi: 0.3389/fonc.2021.742949. eCollection 2021.PMID: 34595121

Round 2

Reviewer 2 Report

No further comments.